# Closing objectivity loophole in Bell tests on a public quantum computer

Adam Bednorz[1],[*] Josep Batle[2,3], Tomasz Białecki[1,4], and Jarosław K. Korbicz[5][†]

[1]*Faculty of Physics, University of Warsaw, ul. Pasteura 5, PL02-093 Warsaw, Poland*
[2]*Departament de Física and Institut d'Aplicacions Computacionals de Codi Comunitari (IAC3),*
*Campus UIB, E-07122 Palma de Mallorca, Balearic Islands, Spain*
[3]*CRISP – Centre de Recerca Independent de sa Pobla, 07420 sa Pobla, Balearic Islands, Spain*
[4]*Faculty of Physics and Applied Informatics, University of Lodz,*
*ul. Pomorska 149/153, PL90-236 Lodz, Poland and*
[5]*Center for Teoretical Physics, Polish Academy of Sciences,*
*al. Lotników 32/46, PL02-668 Warsaw, Poland*

We have constructed and run a Bell test of local realism focusing on the objectivity criterion. The objectivity means that the outcomes are confirmed macroscopically by a few observers at each party. The IBM Quantum and IonQ devices turn out to be sufficiently accurate to pass such an extended Bell-type test, although at the price of communication loopholes and residual but statistically significant signaling. The test also serves as the benchmark of entanglement spread across larger sets of qubits.

## I. INTRODUCTION

Quantum mechanics is incompatible local realism, which is confirmed by violation of Bell inequalities by two separate parties [1–3]. The violation relies on several assumptions, to be satisfied experimentally. First of all, the detection must be complete, every event is recorded and taken into account. In the past, due to low efficiency of photodetectors most of photons got lost, so one applied postselection, reduction of outcomes to the cases where each of photons has clicked [4–7]. It left the detection loophole open, since some local realistic models could exploit low detection rate [8–11]. In modern experiments, photodetection efficiency has increased sufficiently to abandon postselection. Other systems, ions, atoms and transmons are always fully detected so this loophole has never affected them [12–15]. Another challenge is locality, meaning that the two parties are sufficiently separated, so that no exchange of information is possible during a single experiment. When the parties were close to each other, which was the case of atoms, ions and transmons, the timescale was long enough when compared to the distances, and thus communication was in principle possible. To prevent possible communication, one can make a large separation, of the order of tens of meters and more, and make the experiment enough quick [16–20]. The speed of light is taken as the absolute limit of communication, although it is an axiom [21] and current Bell tests are done in some preferred frame, where relativistic invariance is not assumed. One should rather eliminate potential communication channels, like cables and connection, making locality a more contextual than fundamental assumption. The communication loophole depends on freedom of choice. Bell tests assume randomly chosen measurement settings, which must occur locally so that information about the choice of one party does not reach the other before the measurements are accomplished. If one insists on the lack of such freedom, claiming superdeterminism – predetermination of all measurement settings in the past, the communication, although very conspiratory, cannot be excluded. One has to refer to common sense, that the choices are truly and securely random at some stage, by the technical constraints of the setup. Otherwise, no conclusion can be drawn, not only from Bell tests, but majority of experiments relying on some external influence independence. The other side of the communication problem is the end of the measurement. While it is tempting to mark the accomplishment of the readout by a control clock, it is again a matter of common sense agreement. It is better exposed by the Wigner friend problem, where the measurement is performed by an agent (friend) which is observed by another apparatus or human (Wigner) [22–25]. The end of the measurement is marked by the moment when all or the most of external observers receive the outcome. Just like superdeterminism, it becomes a matter of convention or common sense, referring to the technical specification of the detector. The end of the measurement is the moment when the readout outcome becomes objective, visible to many independent observers. One can define quantum objectivity more rigorously by introducing the quantum Darwinism idea [59] and Spectrum Broadcast Structures (SBS) [26, 60], which have been already applied to analysis of quantum measurements [61].

Motivated by the objectivity research, in the hereby paper, we rerun Bell tests on IBMQ and IonQ quantum platforms, simulating 3 observers (friends) at each side, with the requirement that the result $+1$ or $-1$ must be unanimous among the party to be assigned, otherwise we set 0. It builds on the previous tests with 2 friends [28–30] by more friends, more statistics, no postselection, and much more extensive analysis, including no-signaling. The test has been run on IBM Quantum, which is a routine platform for quantum experiments nowadays [31–37], including tests of objectivity [38]. The

---
[*] Adam.Bednorz@fuw.edu.pl
[†] jkorbicz@cft.edu.pl

size of IBM devices is too small compared to measurement time to close the communication loophole. Nevertheless, we have minimized the possibility of communication by (i) separating the parties by a middle qubits, and (ii) applying only phase-dependent settings to avoid crosstalk in energy eigenbasis. Together with the Bell test we also checked no-signaling in the context of IBM Quantum and IonQ limitations. No-signaling means that the setting of one party cannot affect the readout of the other one. In contrast to Bell inequality, it is not allowed both in classical and quantum description [39, 40]. At the fundamental level, signaling would exceed the relativistic information transfer speed limit, namely speed of light. We stress that all performed Bell tests do not directly test relativity as they are set in a preferred frame, and the actual signal speeds through fibers of waveguides are usually smaller that the vacuum limit. Violation of no-signaling indicates at least some unspecified crosstalk, and poses a serious technical hurdle if its causes cannot be identified. In the test with friends, one has to check all 8 combinations of outcomes.

The results show violation of Bell inequality in the stringent version with unanimity condition, but also some violation of no-signaling. The latter is not large in the sense of absolute values, compared to Bell, but still significant, beyond 10 standard deviations. The paper is organized as follows. We start with brief recapitulation of the Bell test, extended to multifriend parties, and definition of no-signaling. Then we explain the actual implementation on IBM Quantum and IonQ. Next, we show and comment the results, and conclude the paper with discussion of interpretation and possible next steps and challenges.

## II. BELL TEST WITH FRIENDS OR THE OBJECTIVITY SCENARIO

The standard Bell test involves two parties, $A$ and $B$, who share an entangled state and make local measurement independent of each other. For the consistency, we shall work in the qubit basis $|0\rangle$ and $|1\rangle$ for each party and assume the entangled state $|\psi\rangle$ of the form

$$\sqrt{2}|\psi\rangle = |00\rangle - i|11\rangle \tag{1}$$

with the local measurement operators

$$A = ie^{i\alpha}|1\rangle\langle 0| - ie^{-i\alpha}|0\rangle\langle 1| \tag{2}$$

and analogously $B$ with $\alpha \to \beta$. The projective measurements in the eigenbasis of $AB$ give outcomes $A, B = \pm 1$ and the correlaction

$$\langle AB \rangle = -\sin(\alpha + \beta). \tag{3}$$

Let each party choose its setting by taking $\alpha_a$, $\beta_b$ for $a, b = 0, 1$, i.e. each party has two independent measurement choices. We shall denote $A_a$ and $B_b$ for these respective choices. For all classical joint probability distributions $P(A_0, A_1, B_0, B_1) \geq 0$, $\sum P = 1$ the Bell-CHSH

inequality holds

$$\mathcal{B} = \langle A_0 B_0 \rangle - \langle A_1 B_0 \rangle - \langle A_0 B_1 \rangle - \langle A_1 B_1 \rangle_{11} \leq 2 \tag{4}$$

with correlations defined as $\langle F \rangle = \sum_{\Omega} P(\Omega) F(\Omega)$. However, in quantum mechanics, setting the angles $\alpha_{0,1} = 0, \pi/2$ and $\beta_{0,1} = -\pi/4, +\pi/4$ we get

$$\langle A_1 B_1 \rangle = \langle A_1 B_0 \rangle = \langle A_0 B_1 \rangle = -\langle A_0 B_0 \rangle = -1/\sqrt{2} \tag{5}$$

which violated (4) at $2\sqrt{2} \simeq 2.828$. It means that the joint probability cannot exist, and the outcomes of the experiment $A_0$, $A_1$, $B_0$, $B_1$ are not a set of counterfactually definite values

The catch in this test is the measurement process, hidden in the projection postulate. In reality, no measurement is perfect, and moreover, it needs some criterion that is has been completed. One reasonable choice is the objectivity of the result - a measurement is completed if the result is objective, accessible to many observers. This requires that the measurement result is stored faithfully in many copies in the macroscopic apparatus and those copies can be accessed by external observers. As an example, one may think of a single pixel of the measuring device, which by emitting or reflecting light can be seen by many observers reading out the measurement result.

Thus, in the friend(s)/objectivity scenario we are simulating here, information is copied to auxiliary systems before measurement. This simulates the information proliferation in the degrees of freedom of the measuring apparatus. Imagine three observers (friends) $A^0$, $A^1$ and $A^2$ with the joint space of states $|A^0 A^1 A^2\rangle$ with $A^i = 0, 1$ and analogously for $B$; see Figs. 2 and 3. We will consider projective measurements on the three qubits,g giving 8 outcomes of the form $A^0 A^1 A^2$ corresponding to projection onto the respective basis states. The initial entanglement is between qubits $A^0$ and $B^0$, i.e. the initial state is

$$\sqrt{2}|\psi\rangle = |000, 000\rangle - i|100, 100\rangle \tag{6}$$

Now the choice of the Bell measurement (2) is realized by a unitary rotation

$$S_\alpha = \frac{1}{\sqrt{2}} \begin{pmatrix} 1 & -ie^{i\alpha} \\ -ie^{-i\alpha} & 1 \end{pmatrix} \tag{7}$$

acting on the qubit $A^0$ in the basis $|0\rangle$, $|1\rangle$ (and $B^0$ with $\beta$). Unlike in the usual Bell scenario, where the state is kept fixed from run to run and the measurements changed, here we use an equivalent approach of changing the local state from run to run but keeping the measurement fixed. This is dictated by the technical capabilities of the simulation platforms used and our desire to distribute as precisely the same information to all observers as possible. Therefore, it is better, error-wise, to set the measurement angles before the information copying rather than angle-dependent coupling individually each observer to the system. Since we want to test

the Bell-type scenario, the rotation must take place before broadcasting information to the friends. The other choice, i.e. individual basis rotation by each observers would lead to a different type of scenario with multipartite entanglement and appropriate generalization of Bell inequalities [41, 42]. At this stage the state reads

$$
\begin{aligned}
\sqrt{8}|\psi_{\alpha\beta}\rangle = {} & (1 + ie^{i(\alpha+\beta)})|000,000\rangle \\
& - (i + e^{-i(\alpha+\beta)})|100,100\rangle \\
& - (ie^{-i\alpha} + e^{i\beta})|100,000\rangle \\
& - (ie^{-i\beta} + e^{i\alpha})|000,100\rangle.
\end{aligned}
\tag{8}
$$

Then we apply the crucial step of information copying. We simulate it via conditional flipping of qubits $A^{1,2}$ if $A^0 = 1$, i.e. we apply a controlled-$X$ gate of the form $CXX|A^0 A^1 A^2\rangle = |A^0 (A^1 \oplus A^0)(A^2 \oplus A^0)\rangle$ with $F \oplus G = (F + G)$ modulo 2; see Fig. 2. In the full basis $|A^0 A^1 A^2\rangle$, this operation reads

$$
\begin{pmatrix}
1 & 0 & 0 & 0 & 0 & 0 & 0 & 0 \\
0 & 1 & 0 & 0 & 0 & 0 & 0 & 0 \\
0 & 0 & 1 & 0 & 0 & 0 & 0 & 0 \\
0 & 0 & 0 & 1 & 0 & 0 & 0 & 0 \\
0 & 0 & 0 & 0 & 0 & 0 & 0 & 1 \\
0 & 0 & 0 & 0 & 0 & 0 & 1 & 0 \\
0 & 0 & 0 & 0 & 0 & 1 & 0 & 0 \\
0 & 0 & 0 & 1 & 0 & 0 & 0 & 0
\end{pmatrix}.
\tag{9}
$$

The state afterwards reads:

$$
\begin{aligned}
\sqrt{8}|\psi_{\alpha\beta}\rangle = {} & (1 + ie^{i(\alpha+\beta)})|000,000\rangle \\
& - (i + e^{-i(\alpha+\beta)})|111,111\rangle \\
& - (ie^{-i\alpha} + e^{i\beta})|111,000\rangle \\
& - (ie^{-i\beta} + e^{i\alpha})|000,111\rangle.
\end{aligned}
\tag{10}
$$

Strictly speaking, from the quantum measurement theory, this not yet the final state of the measurement process as it is obviously entangled between the results 0 and 1. This is the pre-measurement state, which forms after information has been encoded in many degrees of freedom of the measuring apparatus. The final, post-measurement state, is obtained after tracing out some degrees of freedom as unobserved, which in ideal case induces a perfect decoherence and collapse of entanglement into a mixture. However, due to the limited connectivity of the used quantum devices, and hence a limited amount of qubits one can reliably include in the simulation of the measuring apparatus, we will not discard any of the qubits in (10) but rather perform the strict majority vote measurements on each side. We thus require unanimity between observers, i.e.

$$
\begin{aligned}
000 & \to +1, \\
111 & \to -1, \\
\text{otherwise} & \to 0
\end{aligned}
\tag{11}
$$

so the measured observable essentially reads

$$
\bar{A} = |000\rangle\langle000| - |111\rangle\langle111|
\tag{12}
$$

and similarly for $B$. In the ideal case

$$
CXX \, \bar{A} \, CXX = |000\rangle\langle000| - |100\rangle\langle100|
\tag{13}
$$

so the correlations are identical to the Bell test with single observers (4) for $A \to \bar{A}$ and $B \to \bar{B}$. Moreover, only the outcomes 000 and 111 should be registered. Such a test shares the state before measurement. Note that we can do it only after the setting rotation. We cannot clone the state before choosing and applying the $\alpha/\beta$-dependent rotation. Otherwise we would get a highly entangled state

$$
(|000,000\rangle - i|111,111\rangle)/\sqrt{2}
\tag{14}
$$

which gives makes the friends unable to receive identical outcomes. The individual probabilites are given by projective measurements

$$
P(A^0 A^1 A^2, B^0 B^1 B^2) = |\langle A^0 A^1 A^2, B^0 B^1 B^2|\psi_{\alpha\beta}\rangle|^2
\tag{15}
$$

for all values $A^j, B^j = 0, 1$, which gives

$$
\begin{aligned}
P(000, 000) &= P(111, 111) = (1 - \sin(\alpha+\beta))/4, \\
P(000, 111) &= P(000, 111) = (1 + \sin(\alpha+\beta))/4, \\
P(A^0 A^1 A^2, B^0 B^1 B^2) &= 0 \text{ otherwise}
\end{aligned}
\tag{16}
$$

The violation of Bell inequality (4) with the restriction (11) by the state (10) is therefore essentially identical but the result gets a new meaning. It is confirmed collegially by all friends who must be unanimous to count it.

In the test one can simultaneously verify no-signaling, i.e. independence of the result of friends at one party of the setting chosen by the other one. In particular, the probability $P(AB|ab)$ for the settings $a$ and $b$ chosen by the team $A$ and $B$, respectively, gives

$$
\begin{aligned}
\delta P_{a*}(A) &= P(A*|a0) - P(A*|a1) \\
\delta P_{*b}(B) &= P(*B|0b) - P(*B|1b)
\end{aligned}
\tag{17}
$$

with $*$ meaning ignoring the results of the other party. By no-signaling principle $\delta P$ must remain 0 within statistical error. In the ideal case

$$
\begin{aligned}
P(000, ***|a*) &= P(111, ***|a*) = \\
P(***, 000|*b) &= P(***, 111|*b) = 1/2
\end{aligned}
\tag{18}
$$

while all other probabilities are 0. In reality, imperfections make them not zero, while the statistical error is very small so no-signaling can be tested very accurately in this case.

## III. SETUP ON IBM QUANTUM

IBM Quantum provides a set of native gates for single and two-qubit operations. For single qubit rotation we can essentially use (7) which is realize natively.

To create entanglement we have to first apply $S_0$ to a qubit and then use a two qubit coupling, $CX$ (also known as $CNOT$) gate: $CX_\downarrow|FG\rangle = |F(F \oplus G)\rangle$ gate reverses the state $F$ when $G = 1$, while $CX_\uparrow|FG\rangle = |(F \oplus G)G\rangle$ acts in the opposite direction, see details in Appendix A. The pair of $CX$ gates swap the state with $|0\rangle$, i.e.

$$CX_\uparrow CX_\downarrow |\phi 0\rangle = |0\phi\rangle \qquad (19)$$

To create the entangled state we initialize an auxiliary qubit $M$, see Fig. 3, by $|0\rangle \rightarrow S_0|0\rangle = |-y\rangle = (|0\rangle - i|1\rangle)/\sqrt{2}$, and apply $CX_\downarrow$ to $|-_y\ \ 0\rangle$ to get $(|00\rangle - i|11\rangle)\sqrt{2}$ for qubits $M$ and $B^0$. The source $M$ qubit is then swapped to $A^0$ by the pair of opposite $CX$ gates.

The setting are defined by $S_\alpha$ and $S_\beta$ rotations applied to the final qubits $A_0$ and $B_0$, as described in the previous section. The state is then copied to the friends by pair of $CX$ gates, see the full circuit in Fig. 2.

The statistical error can be evaluated for a single shot and the scaled by the number of repetitions, as the accumulation of Bernoulli statistics (Gaussian at large number of repetitions). A single shot gives the variance for correlations

$$\langle \Delta^2(AB)\rangle = \langle A^2 B^2\rangle - \langle AB\rangle^2 \qquad (20)$$

which is $1/2$ in the ideal case. For the test of signaling, we tak the simple Bernoulli formula

$$\langle (\Delta P)^2\rangle = P(1-P) \qquad (21)$$

which is ideally $1/4$ for 000 and 111 while 0 otherwise. In practice, the latter is not zero but still very small, which allows to check no-signaling at very small error.

## IV. RESULTS

We have run the experiment in April 2025 on *ibm_brisbane*, *ibm_sherbrooke* (Eagle generation), on the same groups of qubits, and *ibm_torino*, *ibm_kingston* (Heron generation, revision 1 and 2, respectively), with 60 jobs, 20000 shots and 25 repetitions of each set of settings $ab$, randomly shuffled. so the total number of trials is

$$N = 3 \cdot 10^7 = 30000000 = 25 \cdot 20000 \cdot 60 \qquad (22)$$

The single group is connected as in Fig. 3 and the distribution of the tested 6 groups, see Table I, over the whole network is shown in Fig. 4, with indicated violation of Bell inequality and/or no-signaling. The result of Bell expressions $S$ (4) is given in Table IV, V and Figs. 1, while the test of no-signaling is shown in Figs. 8,9, 10,11. The qubit frequencies for *ibm_brisbane* and *ibm_sherbrooke* are listed in Tables VI, VII, while the other are not disclosed.

The data show that the inequality has been violated by more than 10 standard deviations in the case of 2

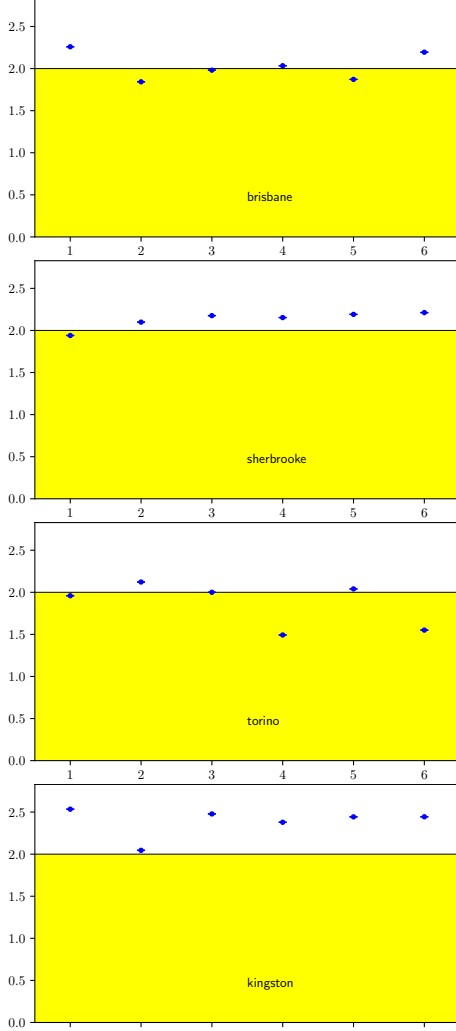

FIG. 1. Bell test results with objectivity condition i.e. all 3 observers at each party must agree on the outcome on *ibm_brisbane*, *ibm_sherbrooke*, *ibm_torino*, *ibm_kingston*.

groups *ibm_brisbane* and 5 groups *ibm_sherbrooke*. Unfortunately, the violation is accompanied by violation of no-signaling in part oth the groups, especially on *ibm_sherbrooke*, which confirms the seriousness of this issue at IBM Quantum [43]. Even acknowledging that the signaling is much smaller than Bell violation and may be related to some crosstalk, we are unable to point out a direct cause. The common ZZ crosstalk, i.e. in the energy eigenspace, is unlikely are the choice operations $S_{\alpha/\beta}$ differ only by phase, not amplitude. There is also no frequency collision [44]. The Heron device, *ibm_torino* shows much smaller signaling. At the newest *ibm_kingston* all groups show violation of Bell inequality and signaling stays within statistical error.

## V.   SETUP ON IONQ

We were able to compare the IBM results with the competitive ion trap technology using IonQ quantum computer, Forte Enterprise 1, with cloud access. It uses ytterbium ions $^{171}\text{Yb}^+$ support two-level qubit space, with the drive frequency 12.64GHz between hyperfine levels, placed in a Paul trap, see details in the documentation [45–51]. Due to all-to-all connectivity the test requires only 6 qubits, with two entangled and the friends. We have run each setting, randomly shuffled, with 10000 shots and 35 repetitions. Unlike IBM, iterating over circuits and then shots, IonQ iterates over shots and then over circuits, which allows in principle some memory. The timescales are much longer, with single qubit gate of $63\mu s$, two-qubit gate $650\mu s$, and readout $250\mu s$, compared to coherence times $\sim 100s$. Errors of single and two-qubit gates are $5 \cdot 10^{-4}$ and $6.6 \cdot 10^{-3}$ respectively. The circuit is depicted in Fig. 7. and the details in Appendix A. The result of Bell combination is $2.569\pm0.0024$ which is undoubtedly violating the classical bound 2. There is also significant violation of no-signaling, Fig. 12, but further tests are necessary to check if it is accidental.

The full data and scripts are publicly available [52].

## VI.   DISCUSSION

The Bell inequality strengthened by the objective friends is in general sometimes violated at a quantum processor. The fact that only some cases show the violation which is anyway far below the quantum ideal value indicates problematic error propagation. It calls for further research in order to identify the origins and countermeasures. It seems that the newest generation of IBM Heron, revision 2, represented by *ibm_kingston* has overcome some problems of earlier devices, with all groups passing the test. Anyway, the successful violations give the confidence in entanglement as a nonclassical feature and sufficient quality to spread entanglement on the medium-range distances. Further improved tests should set the limit on distances and the number of friends able to violate the inequality. Of the serious concern is the violation of no-signaling, which indicates at least serious technical failures. Resolving such problems is necessary for the progress in reliable quantum computation.

### ACKNOWLEDGMENTS

JKK acknowledges the financial support of the National Science Centre (NCN) through the QuantEra project Qucabose 2023/05/Y/ST2/00139. We acknowledge the use of IBM Quantum services for experiments in this paper. The views expressed are those of the authors, and do not reflect the official policy or position of IBM or the IBM Quantum team.

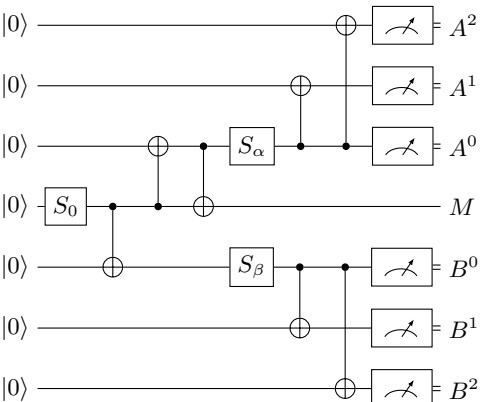

FIG. 2. The circuit of the Bell test with friends. The $CX$ gates are denoted by the lin between control (depicted as ●) and target (depicted as ⊕). The operational flow is from the left to the right.

FIG. 3. The connections in the objectivity test. the middle (yellow) qubit $M$ lies between the two parties $A$ and $B$ (green and red respectively) the main qubits $A^0$ and $B^0$ are coupled to the friends.

**Appendix A: Implementation of necessary gates**

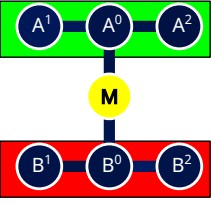

FIG. 13. The notation of the ECR gate in the convention $ECR_\downarrow|FG\rangle$

| G | $A^2$ | $A^1$ | $A^0$ | $M$ | $B^0$ | $B^1$ | $B^2$ |
|---|---|---|---|---|---|---|---|
| 1 | 3 | 5 | 4 | 15 | 22 | 21 | 23 |
| 2 | 27 | 29 | 28 | 35 | 47 | 46 | 48 |
| 3 | 40 | 42 | 41 | 53 | 60 | 59 | 61 |
| 4 | 65 | 67 | 66 | 73 | 85 | 84 | 88 |
| 5 | 78 | 80 | 79 | 91 | 98 | 97 | 99 |
| 6 | 103 | 105 | 104 | 111 | 122 | 121 | 123 |

TABLE I. Groups qubits used in the demonstration on *ibm_brisbane* and *ibm_sherbrooke*

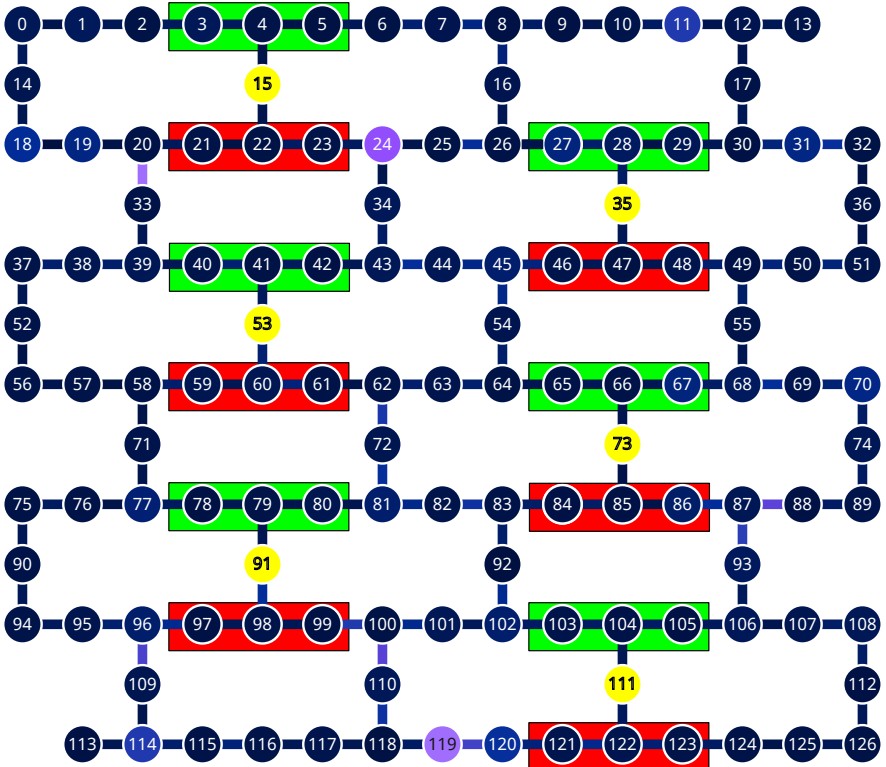

FIG. 4. The placement of all the tested groups on the *ibm_brisbane* and *ibm_sherbrooke*.

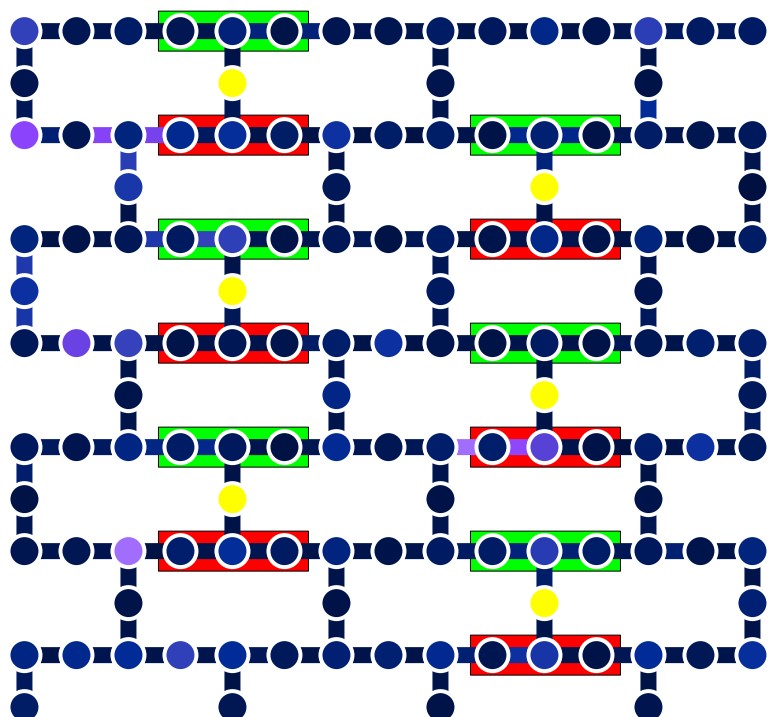

FIG. 5. The placement of all the tested groups on the *ibm_torino*.

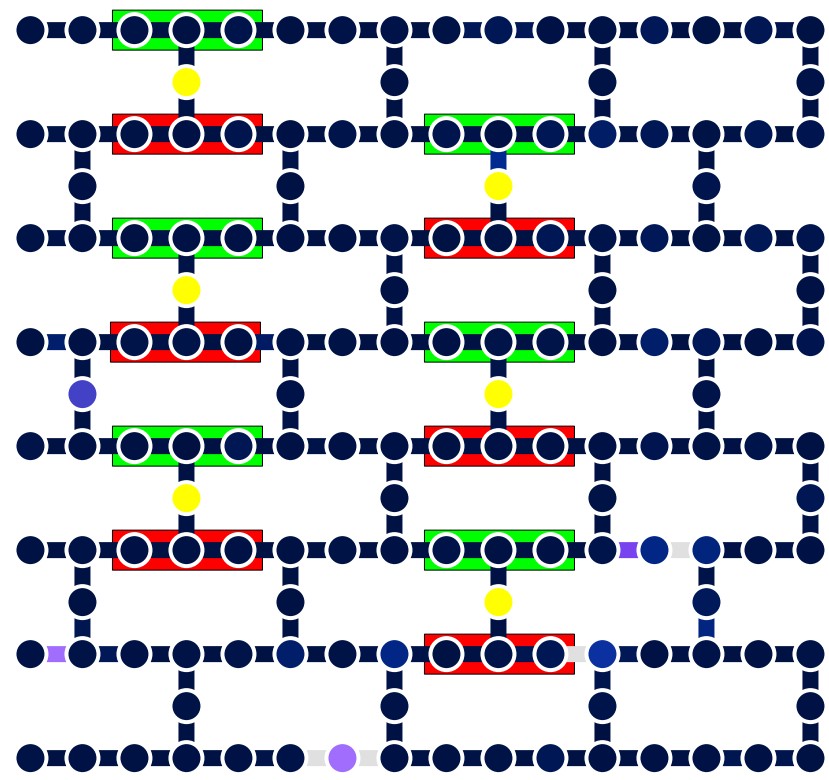

FIG. 6. The placement of all the tested groups on the *ibm_kingston*.

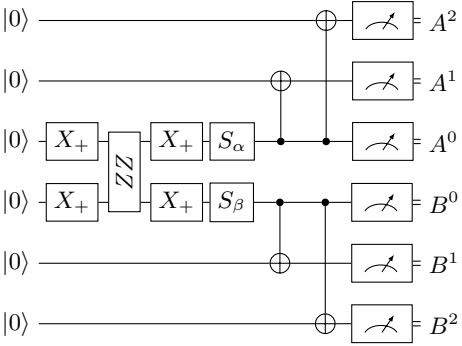

FIG. 7. The circuit of the Bell test with friends on IonQ. The operational flow is from the left to the right.

To run our test we require $CX$ gate,

$$CX_\downarrow = \begin{pmatrix} 1 & 0 & 0 & 0 \\ 0 & 1 & 0 & 0 \\ 0 & 0 & 0 & 1 \\ 0 & 0 & 1 & 0 \end{pmatrix} \tag{A1}$$

in the basis $|00\rangle$, $|01\rangle$, $|10\rangle$, $|11\rangle$. We denote reversed gates by $\langle F'G'|U_\uparrow|FG\rangle = \langle G'F'|U_\downarrow|GF\rangle$. The ideal $CX$ gate is its own reverse, real, and Hermitian. The $CX$ gates are actually transpiled by echo cross-resonance gates ($ECR$); see Appendix.

For the physical implementation and manipulation of qubits as transmons [53] and programming of IBM Quan-

| G | $A^2$ | $A^1$ | $A^0$ | $M$ | $B^0$ | $B^1$ | $B^2$ |
|---|---|---|---|---|---|---|---|
| 1 | 3 | 5 | 4 | 16 | 23 | 22 | 24 |
| 2 | 28 | 30 | 29 | 36 | 48 | 47 | 49 |
| 3 | 41 | 43 | 42 | 54 | 61 | 60 | 62 |
| 4 | 66 | 68 | 67 | 74 | 86 | 85 | 87 |
| 5 | 79 | 81 | 80 | 92 | 99 | 98 | 100 |
| 6 | 104 | 106 | 105 | 112 | 124 | 123 | 125 |

TABLE II. Groups qubits used in the demonstration on *ibm_torino*

| G | $A^2$ | $A^1$ | $A^0$ | $M$ | $B^0$ | $B^1$ | $B^2$ |
|---|---|---|---|---|---|---|---|
| 1 | 2 | 4 | 3 | 16 | 23 | 22 | 24 |
| 2 | 28 | 30 | 29 | 38 | 49 | 48 | 50 |
| 3 | 42 | 44 | 43 | 56 | 63 | 62 | 64 |
| 4 | 68 | 70 | 69 | 78 | 89 | 88 | 90 |
| 5 | 82 | 84 | 83 | 96 | 103 | 102 | 104 |
| 6 | 108 | 110 | 109 | 118 | 129 | 128 | 130 |

TABLE III. Groups qubits used in the demonstration on *ibm_kingston*

tum, see [54–56]. The IBM Quantum devices use a native asymmetric two-qubit $ECR$ gates [57, 58] instead of $CX$ but one can transpile the latter by the former, adding single qubits gates We shall use Pauli matrices in the

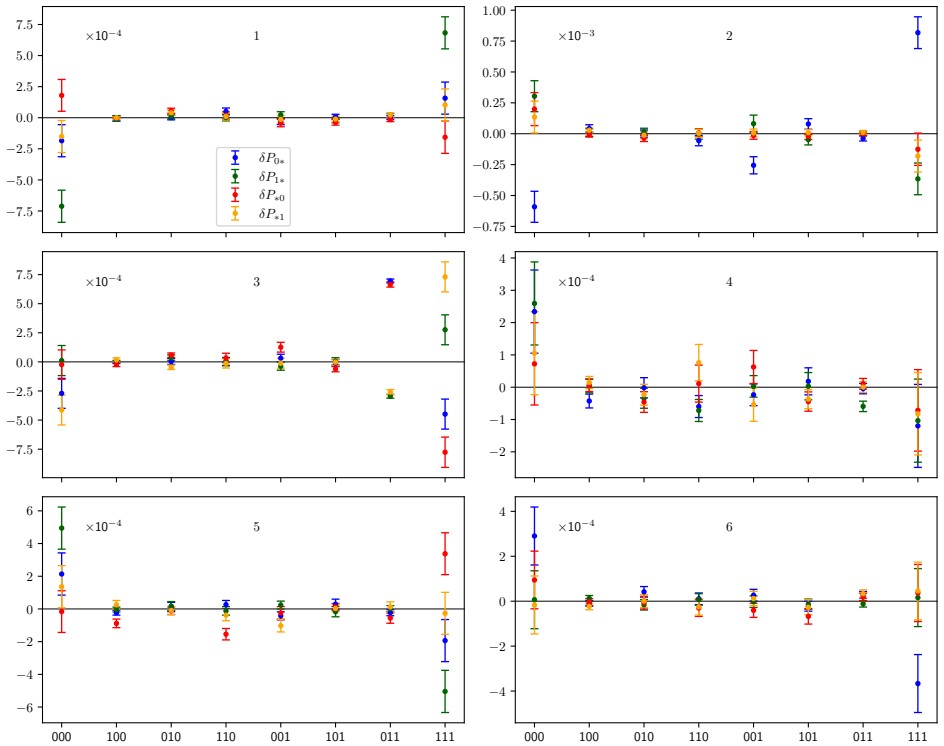

FIG. 8. Signaling between qubits $\delta P_{a*}(A^1 A^2 A^0)$ and $\delta P_{*b}(B^1 B^2 B^0)$ in each of the groups on *ibm_brisbane*.

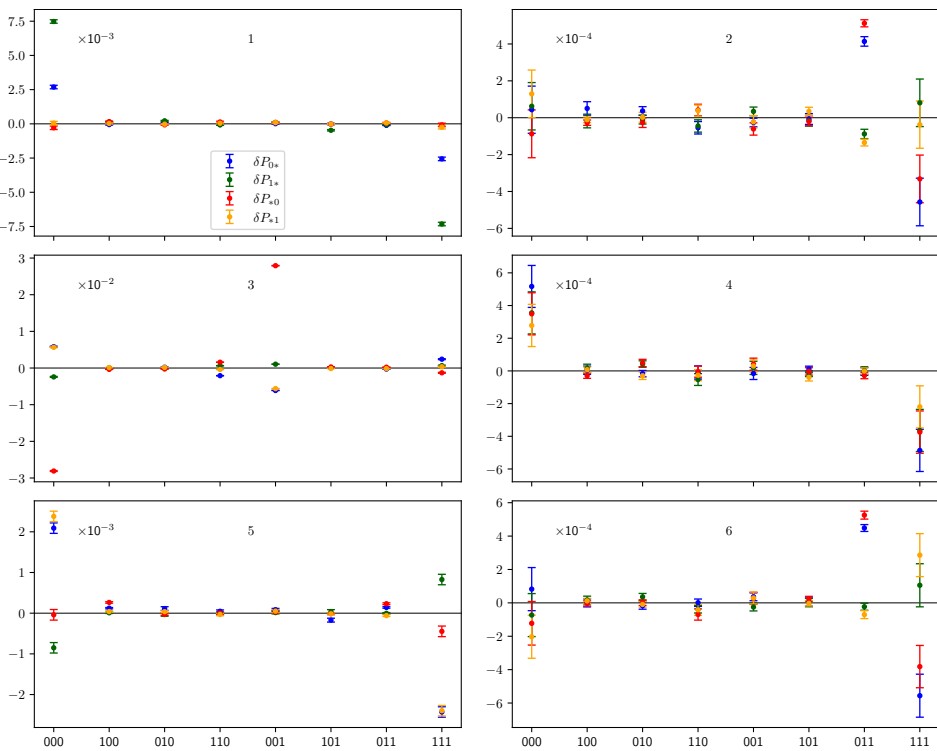

FIG. 9. Signaling between qubits $\delta P_{a*}(A^1 A^2 A^0)$ and $\delta P_{*b}(B^1 B^2 B^0)$ in each of the groups on *ibm_sherbrooke*.

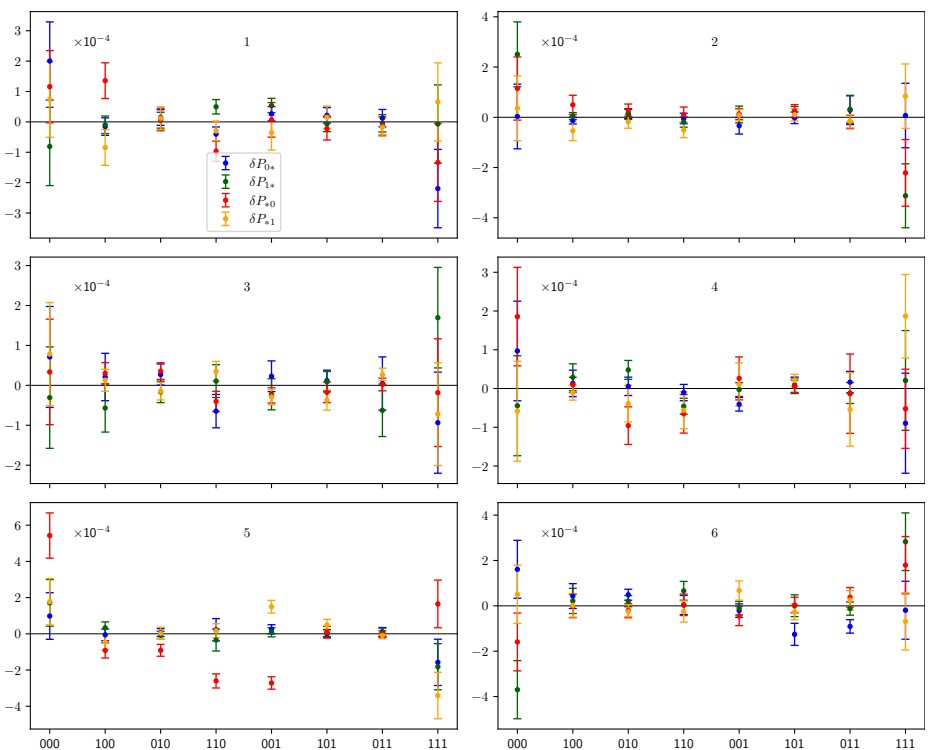

FIG. 10. Signaling between qubits $\delta P_{a*}(A^1 A^2 A^0)$ and $\delta P_{*b}(B^1 B^2 B^0)$ in each of the groups on *ibm_torino*.

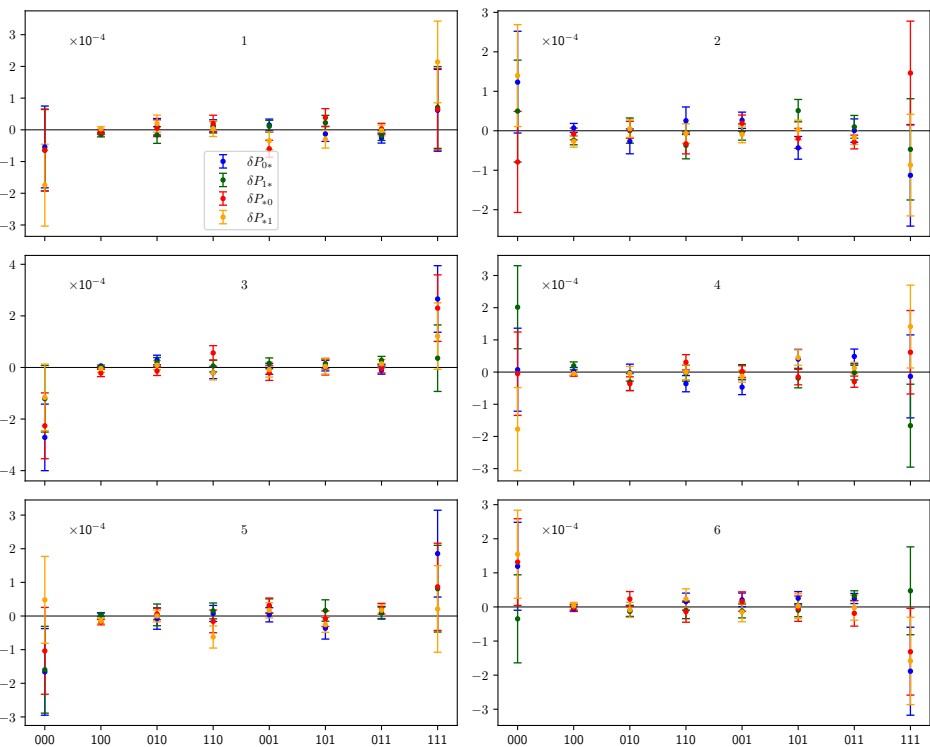

FIG. 11. Signaling between qubits $\delta P_{a*}(A^1 A^2 A^0)$ and $\delta P_{*b}(B^1 B^2 B^0)$ in each of the groups on *ibm_kingston*.

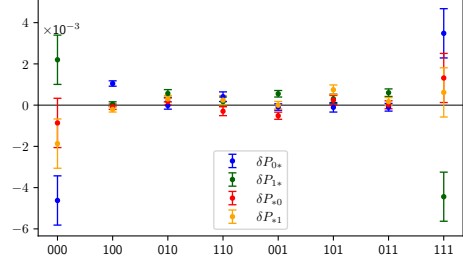

FIG. 12. Signaling between qubits $\delta P_{a*}(A^1 A^2 A^0)$ and $\delta P_{*b}(B^1 B^2 B^0)$ in each of the groups on IonQ Forte Enterprise 1.

| G | $\mathcal{B}_b$ | $\Delta\mathcal{B}_b[10^{-4}]$ | $A-B$ | $\mathcal{B}_s$ | $\Delta\mathcal{B}_s[10^{-4}]$ | $A-B$ |
|---|---|---|---|---|---|---|
| 1 | 2.259 | 2.75 | $\leftarrow$ | 1.940 | 2.62 | $\leftarrow$ |
| 2 | 1.843 | 2.69 | $\leftarrow$ | 2.099 | 2.73 | $\leftrightarrow$ |
| 3 | 1.981 | 2.74 | $\leftrightarrow$ | 2.176 | 2.62 | $\leftrightarrow$ |
| 4 | 2.034 | 2.67 | | 2.153 | 2.73 | |
| 5 | 1.872 | 2.94 | | 2.191 | 2.67 | $\leftrightarrow$ |
| 6 | 2.194 | 2.79 | | 2.212 | 2.81 | $\leftrightarrow$ |

TABLE IV. Results of Bell tests $\mathcal{B}_{b/s}$ (4) on each group with the error $\Delta\mathcal{B}_{b/s}$ for *ibm_brisbane/ibm_sherbrooke*, respectively, for each group of qubits. The significant violation of no signaling is marked by $A \rightarrow B$, $A \leftarrow B$, $A \leftrightarrow B$, for respective direction of signaling. The criterion of significance is the nonzero value beyond 5 standard deviations

| G | $\mathcal{B}_t$ | $\Delta\mathcal{B}_t[10^{-4}]$ | $A-B$ | $\mathcal{B}_k$ | $\Delta\mathcal{B}_k[10^{-4}]$ | $A-B$ |
|---|---|---|---|---|---|---|
| 1 | 1.959 | 2.66 | | 2.535 | 2.62 | |
| 2 | 2.122 | 2.74 | | 2.045 | 2.82 | |
| 3 | 2.000 | 2.62 | | 2.478 | 2.68 | |
| 4 | 1.493 | 2.64 | | 2.379 | 2.72 | |
| 5 | 2.040 | 2.68 | $\leftarrow$ | 2.443 | 2.67 | |
| 6 | 1.551 | 2.74 | | 2.444 | 2.63 | |

TABLE V. Results of Bell tests $\mathcal{B}_{t/k}$ (4) on each group with the error $\Delta\mathcal{B}_{t/k}$ for *ibm_torino/ibm_kingston*, respectively, for each group of qubits. The significant violation of no signaling is marked by $A \rightarrow B$, $A \leftarrow B$, $A \leftrightarrow B$, for respective direction of signaling. The criterion of significance is the nonzero value beyond 5 standard deviations

basis $|0\rangle$, $|1\rangle$,

$$X = \begin{pmatrix} 0 & 1 \\ 1 & 0 \end{pmatrix}, Y = \begin{pmatrix} 0 & -i \\ i & 0 \end{pmatrix}, Z = \begin{pmatrix} 1 & 0 \\ 0 & -1 \end{pmatrix}, I = \begin{pmatrix} 1 & 0 \\ 0 & 1 \end{pmatrix}. \tag{A2}$$

The $ECR$ gate acts on the states $|FG\rangle$ as (Fig. 13)

$$ECR_\downarrow = (XI - YX)/\sqrt{2} = CR^-(XI)CR^+ =$$

$$\begin{pmatrix} 0 & X_- \\ X_+ & 0 \end{pmatrix} = \begin{pmatrix} 0 & 0 & 1 & i \\ 0 & 0 & i & 1 \\ 1 & -i & 0 & 0 \\ -i & 1 & 0 & 0 \end{pmatrix}/\sqrt{2}, \tag{A3}$$

in the basis $|00\rangle$, $|01\rangle$, $|10\rangle$, $|11\rangle$ where the native gate is

$$X_+ = X_{\pi/2} = (I - iX)/\sqrt{2} = \begin{pmatrix} 1 & -i \\ -i & 1 \end{pmatrix}/\sqrt{2}, \tag{A4}$$

| G | $A_2$ | $A_1$ | $A_0$ | $M$ | $B_0$ | $B_1$ | $B_2$ |
|---|---|---|---|---|---|---|---|
| 1 | 4875 | 4734 | 4818 | 4947 | 5038 | 4966 | 4844 |
| 2 | 4747 | 4628 | 4994 | 4907 | 4769 | 4853 | 4844 |
| 3 | 4868 | 5063 | 4935 | 4996 | 4766 | 4971 | 4793 |
| 4 | 4957 | 5113 | 4885 | 4979 | 5104 | 4679 | 4898 |
| 5 | 4643 | 5031 | 4863 | 4914 | 4942 | 4816 | 5060 |
| 6 | 4933 | 4982 | 4765 | 4853 | 4937 | 4967 | 5034 |

TABLE VI. Drive frequencies [MHz] of qubits used in the demonstration on *ibm_brisbane*

| G | $A_2$ | $A_1$ | $A_0$ | $M$ | $B_0$ | $B_1$ | $B_2$ |
|---|---|---|---|---|---|---|---|
| 1 | 4747 | 4850 | 4787 | 4599 | 4668 | 4773 | 4758 |
| 2 | 4680 | 4791 | 4742 | 4914 | 4795 | 4674 | 4707 |
| 3 | 4705 | 4654 | 4814 | 4767 | 4672 | 4810 | 4901 |
| 4 | 4758 | 4893 | 4809 | 4874 | 4745 | 4666 | 4889 |
| 5 | 4861 | 5042 | 4786 | 4893 | 4774 | 4949 | 4832 |
| 6 | 4694 | 4883 | 4783 | 4899 | 4731 | 4847 | 4820 |

TABLE VII. Drive frequencies [MHz] of qubits used in the demonstration on *ibm_sherbrooke*

and $X_- = X_{-\pi/2} = ZX_+Z$, with

$$CR^\pm = (ZX)_{\pm\pi/4}, \tag{A5}$$

using the convention $V_\theta = \exp(-i\theta V/2) = \cos(\theta/2) - iV\sin(\theta/2)$ if $V^2 = I$ or $II$. The gate is its inverse, i.e. $ECR_\downarrow ECR_\downarrow = II$.

Note that $Z_\theta = \exp(-i\theta Z/2) = \mathrm{diag}(e^{-i\theta/2}, e^{i\theta/2})$ is a virtual gate adding essentially the phase shift to next gates. The actually performed operations have always the form $Z_{-\theta}UZ_\theta$ where $U$ is some basis gate. $ECR$ is asymmetric, i.e. the qubits are not interchangeable, but it can be reversed, i.e., for $F \leftrightarrow G$, (Fig. 14)

$$ECR_\uparrow = (IX - XY)/\sqrt{2} = (HH)ECR_\downarrow(Y_+Y_-), \tag{A6}$$

denoting $V_\pm = V_{\pm\pi/2}$, and Hadamard gate,

$$H = (Z + X)/\sqrt{2} = Z_+X_+Z_+ = \begin{pmatrix} 1 & 1 \\ 1 & -1 \end{pmatrix}/\sqrt{2}, \tag{A7}$$

and $Z_\pm X_+ Z_\mp = Y_\pm$, with $Y_+ = HZ$ and $Y_- = ZH$.

The $CNOT$ gate can be expressed by $ECR$ (Fig. 15)

$$CX_\downarrow = (II + ZI + IX - ZX)/2 =$$

$$\begin{pmatrix} I & 0 \\ 0 & X \end{pmatrix} = \begin{pmatrix} 1 & 0 & 0 & 0 \\ 0 & 1 & 0 & 0 \\ 0 & 0 & 0 & 1 \\ 0 & 0 & 1 & 0 \end{pmatrix}$$

$$= (Z_+ I) ECR_\downarrow (XS), \tag{A8}$$

while its reverse reads (Fig. 16)

$$CX_\uparrow = (II + IZ + XI - XZ)/2 =$$

$$\begin{pmatrix} 1 & 0 & 0 & 0 \\ 0 & 0 & 0 & 1 \\ 0 & 0 & 1 & 0 \\ 0 & 1 & 0 & 0 \end{pmatrix} = (HH) X_\downarrow (HH)$$

$$= (HH) ECR_\downarrow (SS)(Z_- H). \tag{A9}$$

IBM Heron devices have native $CZ$ gates, i.e. $CZ|ab\rangle = (-1)^{ab}|ab\rangle$ (only flips the sign of the $|11\rangle$ state) and $CX_\downarrow = (IH)CZ(IH)$.

IonQ Forte native entangling gate is $(ZZ)_{\pi/4} = \exp(-iZZ\pi/4)$ that reads

$$\begin{pmatrix} 1 & 0 & 0 & 0 \\ 0 & i & 0 & 0 \\ 0 & 0 & i & 0 \\ 0 & 0 & 0 & 1 \end{pmatrix} \tag{A10}$$

whis is symmetric and can express $CX$ gate as

$$(Z_+ Z_{\pi/4})(IY_+)(ZZ)_{\pi/4}(IX_-)(IZ_{\pi/4}) \tag{A11}$$

or equivalently, see Fig. 17.

FIG. 14. The $ECR_\uparrow$ gate expressed by $ECR_\downarrow$

FIG. 15. The $CX_\downarrow$ gate expressed by $ECR_\downarrow$

FIG. 16. The $CX_\uparrow$ gate expressed by $ECR_\downarrow$

FIG. 17. The $CX_\downarrow$ gate expressed by $ZZ$ gates on IonQ. The outer $Z$ rotations are actually dropped as they do not change the state and measurement.

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
