# Peer review of "Closing objectivity loophole in Bell tests on a public quantum computer"

_SciPost Physics_

## Round 2 · Referee Report · Anonymous (Referee 1) · 2025-9-29

Strengths

In the manuscript titled “Closing objectivity loophole in Bell tests on a public quantum computer”, the authors describe a Bell test run on publicly available IBM Quantum and IonQ devices, with an aim to close an “objectivity” loophole. The usual loopholes considered in Bell tests concern the detection efficiency, locality and freedom-of-choice. Here on the other hand, the focus is on the criterion of objective readout, i.e., the cloneability of the measurement outcome at each of the two sites in a CHSH test. To this end, the authors introduce two additional qubits per site which act as observers/friends, and describe a procedure whereby a majority of observers at a site have to agree upon the measurement outcome before it is declared objective. In this process, the end of the measurement occurs at the moment when the readout outcome becomes objective, agreed upon by multiple independent observers.

1) The proposed modification (and implementation) to existing Bell test on public quantum computers is relatively simple.

2) The paper is written clearly (modulo some typos and lack of clear definitions).

3) The paper would be of interest to researchers in quantum foundations.

Weaknesses

Having said that, 1) It is questionable how novel the introduced method is,

2) How far the proposal goes towards closing aforesaid loophole, and whether the task cannot be achieved by simpler means.

3) It is also unclear how important such a loophole is and whether closing it has any practical relevance say for device-independent cryptography or tests of quantum entanglement.

As such, in my opinion, the paper is not suitable for publication in a prestigious journal such as SciPost Physics in its present form.

Report

1) The objectivity loophole requires a clear definition and a discussion on its importance. The idea would be that an unspecified sub-luminal interaction (or an adversary in a device-independent scenario) could, in principle, change Alice’s registered value after the readout time tag based on the information about Bob’s input choice. In this sense, the violation of the CHSH inequality would be explainable by an LHV model supplemented with communication since the final time tag would already be within the lightcone. But then, a simpler (classical) solution would be to simply copy the readout of Alice’s measurement to an independent classical memory along with the final time tag and store both copies. It would be helpful if the authors could discuss why their method is a preferred solution to closing the objectivity loophole.

It is also unclear how good the two additional qubits per site are as observers in such a fundamental (aiming to be loophole-free) Bell test. For instance, wouldn’t the proposal in the paper require Alice and Bob to trust the quantum nature of their device (i.e., that it implements specific control-NOT operations and majority vote measurements). Closing the objectivity loophole in the present setup would require a more involved analysis to show that no possibility exists for an LHV model supplemented with sub-luminal communication or an adversary in a DI protocol to simulate the observed statistics.

2) A second issue (related to comment 1 above) that needs clarification is the issue of apparent signalling. The presence of apparent signalling seems to undermine the attempt to close the objectivity loophole, some clear analysis is needed to explain how the objectivity loophole has been closed to statistical significant levels despite the apparent signalling. Some results such as Quantum 9, 1760 (2025), Nat. Comm. 16, 4390 (2025), Phil. Trans. R. Soc. A.38220230011 (2024) may be useful in this regard.

3) It was also unclear if the authors are post-selecting on outcomes 000 and 111, and ignoring rounds in which other outcomes were observed. It would be helpful to clarify this.

4) In Reference [24], Nature Physics 16, 1199 (2020), Bong et al. introduced an Extended Wigner's Friend Scenario using a photonic qubit to play the role of each of the ``friends'' and measured violations of a generalisation of Bell’s inequality that they term “Local Friendliness” inequalities. These inequalities are derived based on the assumptions: (i) “absoluteness of observed events” (every observed event happens for all observers), (ii) freedom-of-choice (free choices can be made uncorrelated with other events outside their future light cone) and (iii) locality (the probability of an event is unchanged by conditioning on a free choice made at a space like separated location). There again, a photonic qubit is taken to be a physical system that counts as an “observer”. In principle, the experiment in that paper could also be interpreted as a Bell test with additional observers. The novelty in the current paper of considering 3 observers instead of 2 does not seem to be of major significance.

It would be helpful to include a discussion on the relationship between the present manuscript and the literature, especially with regards to the Local Friendliness inequalities. For instance, couldn’t the authors claim a stronger result by testing for a violation of the LF inequalities instead of the CHSH Bell inequality in the present paper (i.e., relaxing the assumption of absoluteness of observed events in the LF inequalities)?

5) A number of typos in the manuscript need fixing: p1 column 1 “Quantum mechanics is incompatible with local realism”, p2 c2 “it needs some criterion that is has been completed”, p3 c2 “which gives makes the friends unable to receive identical outcomes”, p4 c4 “violation of no-signaling in part of the groups”, etc. and many more which can be improved in a revised version by a careful reading of the manuscript.

Requested changes

See report above.

Recommendation

Ask for major revision

---

## Round 2 · Referee Report · Anonymous (Referee 2) · 2025-10-4

Disclosure of Generative AI use

The referee discloses that the following generative AI tools have been used in the preparation of this report:

I used ChatGPT solely as a writing assistant, to refine the language and improve clarity of my comments. All ideas, analyses, and technical content are my own

Strengths

please see attached file

Weaknesses

please see attached file

Report

please see attached file

Attachment

Recommendation

Reject

---

## Editorial Decision

resubmitted